# Novel NMR Assignment Strategy Reveals Structural Heterogeneity in Solution of the nsP3 HVD Domain of Venezuelan Equine Encephalitis Virus

**DOI:** 10.3390/molecules25245824

**Published:** 2020-12-10

**Authors:** Peter Agback, Andrey Shernyukov, Francisco Dominguez, Tatiana Agback, Elena I. Frolova

**Affiliations:** 1Department of Molecular Sciences, Swedish University of Agricultural Sciences, P.O. Box 7015, SE-750 07 Uppsala, Sweden; andreysh@nioch.nsc.ru (A.S.); tatiana.agback@slu.se (T.A.); 2Laboratory of Magnetic Radiospectroscopy, N.N. Vorozhtsov Institute of Organic Chemistry, SB RAS, 630090 Novosibirsk, Russia; 3Department of Microbiology, University of Alabama at Birmingham, Birmingham, AL 35294-2170, USA; fran81@uab.edu (F.D.); efrolova@uab.edu (E.I.F.)

**Keywords:** NMR, MUSIC pulse sequences, nsP3, IDP, VEEV

## Abstract

In recent years, intrinsically disordered proteins (IDPs) and disordered domains have attracted great attention. Many of them contain linear motifs that mediate interactions with other factors during formation of multicomponent protein complexes. NMR spectrometry is a valuable tool for characterizing this type of interactions on both amino acid (aa) and atomic levels. Alphaviruses encode a nonstructural protein nsP3, which drives viral replication complex assembly. nsP3 proteins contain over 200-aa-long hypervariable domains (HVDs), which exhibits no homology between different alphavirus species, are predicted to be intrinsically disordered and appear to be critical for alphavirus adaptation to different cells. Previously, we have shown that nsP3 HVD of chikungunya virus (CHIKV) is completely disordered with low tendency to form secondary structures in free form. In this new study, we used novel NMR approaches to assign the spectra for the nsP3 HVD of Venezuelan equine encephalitis virus (VEEV). The HVDs of CHIKV and VEEV have no homology but are both involved in replication complex assembly and function. We have found that VEEV nsP3 HVD is also mostly disordered but contains a short stable α-helix in its C-terminal fragment, which mediates interaction with the members of cellular Fragile X syndrome protein family. Our NMR data also suggest that VEEV HVD has several regions with tendency to form secondary structures.

## 1. Introduction

Although IDPs are highly dynamic leading to the narrowing of the linewidth of the resonances in the NMR spectra vs. well-structured proteins with similar molecular weight, they are best described as heterogeneous structural ensembles [1,2,3]. IDPs often contain structural heterogeneities and transiently folded regions, such as transiently populated secondary or tertiary structures, long-range interactions, or aggregation. Notwithstanding, these transiently structured regions of IDP are of particular interest in the study of the biological function of IDPs. They can, however, be responsible for unfavorable relaxation properties and lead to poor spectral quality.

Complete backbone and side chain resonance assignment of NMR spectra of large IDPs is still a challenge due to (a) the reduced frequency dispersion observed in the NMR spectra, especially in the ^1^H dimension and (b) a high degree of divergence in the conformational flexibility characteristic of an IDP. Currently, complete assignment of the NMR resonances demands considerable NMR time to collect the data and extensive human resources to analyze it. The recent success in the newly developed 13C observed experiments [4,5,6,7,8] helps to overcome the problem of the reduced frequency dispersion. However, one should consider the higher sensitivity of the ^1^H nucleus compared to the ^13^C nucleus and the need to assign the amide proton, which are critically important, particularly in ligand titration experiments. This makes the search for additional effective and robust alternatives for ^1^H(N)-based resonance assignment experiments of IDPs urgent. Most of the currently available approaches are better applicable to fully folded proteins (FP) and protocols, which allow the generation of more reliable outcomes of assignment of IDPs that are not yet available. This is because for IDPs, the chemical shifts (CS) of the nuclei are, in many cases, the only source for obtaining the experimental information and thus the structural analysis.

As mentioned above, the time-consuming process of acquisition of NMR data and assignment is a limitation to the use of NMR spectroscopy in the studies of IDPs. This significantly reduces the pace of the workflow and slows down the studies, particularly, if they include numerous IDPs. In this study, we exploited the advantage of MUSIC (MUltiplicity Selective In-phase Coherence transfer, i.e., amino-acid-type edited) methodology [9] to generate conventionally looking 2D ^1^H-^15^N-HSQC spectra. The latter spectra show not only one or two selectively excited types of amino acids, but also the next sequential amides. We adopted and modified the MUSIC pulse sequences (Bruker pulse sequence library (TopSpin3.5)) in a number of ways. (1) We converted the evolution period for the indirect ^15^N dimension from constant time (CT) to semiconstant time (semi-CT) evolution in the pulse sequences. This enabled us to obtain maximum spectral resolution controlled by only the line-widths of the IDPs and make full use of nonuniform sampling (NUS) [10]. (2) Implementation of TROSY-based NMR methods [11] in the MUSIC type pulse sequences made the experiments on larger IDPs possible and allowed more effective use of higher field spectrometers. (3) In addition, the MUSIC pulse sequences were adopted with sensitivity-enhanced options [12].

Another feature of IDPs is that they are, in many cases, rich in proline residues. Lacking the amide proton, prolines make breaks in the sequential assignment procedure. However, there are MUSIC type experiments giving Proline-1 and Proline + 1 correlations, and by combining those experiments with our abovementioned modifications, prolines are no longer a problem in the assignment procedure.

In this study, we also demonstrated the applicability and simplicity of the combination of advanced NMR acquisition and assignment procedures combined with chemical denaturation titration (CDT)-NMR protocols [13]. The chemical denaturants disrupt any residual structure and the resulting rapid exchange between conformers in IDP ensembles provided better quality relaxation properties and spectra.

For the proof-of-principle of this approach, we used a hypervariable domain of the nonstructural protein 3 (nsP3) of Venezuelan equine encephalitis virus (VEEV). VEEV is a representative member of the large *Alphavirus* genus in the *Togaviridae* family [14]. It is one of the most pathogenic alphaviruses, which causes in humans severe meningoencephalitis with frequent neurological sequelae.

Alphavirus nsP3 proteins play critical roles in the early steps of viral RNA replication. Their long, intrinsically disordered hypervariable domains (HVDs) function as hubs for assembly of RNA replication complexes [15,16,17,18]. They are predicted to be intrinsically disordered and recruit multiple host factors, which play indispensable role(s) in RNA replication. Evolution of HVDs mediates alphavirus adaptation to different vertebrate hosts and mosquito vectors. Therefore, structural studies of alphavirus HVDs, and VEEV HVD in particular, are critical for our understanding of viral replication complex assembly and function on molecular level. They will ultimately lead to development of therapeutic means for treatment of highly debilitating, alphavirus diseases. Full assignment of VEEV HVD is a first step in understanding of its function on molecular level.

In this report, we applied a combination of above-described approaches to generate a comprehensive assignment of ^1^H, ^13^C, and ^15^N resonances of the VEEV HVD. This will ultimately allow us to identify the residues that are crucial for ligand binding and involved in development of binding-associated conformational changes in HVD. Our on-going ligand titration experiments will reveal whether VEEV adopts binding mode similar to that we described for chikungunya virus (CHIKV) HVD. In the future, this will allow us to identify the characteristics of HVD that are common for other alphaviruses.

## 2. Materials and Methods

### 2.1. Protein Purification

The codon-optimized sequences of VEEV nsP3 HVD-coding sequence (aa 331–515) was cloned into pE-SUMOpro-3 plasmid (LifeSensors Inc., Malvern, PA, USA) between Nco I and Xho I restriction sites. The resulting sequence encoded N terminal His-Taq with a few extra amino acids (MGHHHHHHGSLW) and lacked last 35 aa of natural HVD, which contained the second C-terminal repeating element. The presence of long, almost identical repeats complicated NMR spectra assignment. Importantly, a VEEV variant lacking one of the repeats still efficiently replicates [15,18]. The aa sequence of HVD with corresponding numbering is presented in Figure 2. The resulting plasmid was used to transform the *E. coli* strain LEMO21(DE3) (New England Biolabs, Ipswich, MA, USA). The ^15^N- and ^15^N^13^C-labelled proteins were produced by growing bacteria in M9 minimal media supplemented with (^15^N)NH_4_Cl and d-(^13^C_6_)glucose (Cambridge Isotope Laboratories, Tewksbury, MA, USA). Protein expression was induced by 1 mM isopropyl β-d-1-thiogalactopyranoside (IPTG) after cell reached the density of ~2 OD_600_. Cells continued to grow for 3 h at 37 °C. Freshly prepared or frozen pellets were lysed in Emulsiflex B15 (Avestin). The recombinant VEEV HVD was purified on a HisTrap HP column (GE Healthcare, Chicago, IL, USA). Size exclusion chromatography on Superdex 200 10/30 column (GE Healthcare) in phosphate buffer (50 mM N_2_HPO_4_ pH 6.8, 200 mM NaCl, 1 mM tris(hydroxypropyl)phosphine (TCEP)) was used as a final purification step. The purified VEEV HVD was exchanged into final NMR buffer (50 mM N_2_HPO_4_ pH 6.8, 50 mM NaCl, 2 mM TCEP and 1 mM NaN_3_) and concentrated to ~0.85 mM. The protein purity and identity were confirmed by SDS-PAGE and mass spectrometry, respectively. Protein concentrations were determined on 280 nm using an extinction coefficient of 12,660 M^–1^ cm^–1^, which was determined using ProteinCalculator v3.4 (San Diego, CA, USA, http://protcalc.sourceforge.net/). For the simplicity, we will call VEEV HVD as vHVD in the paper.

### 2.2. NMR Spectroscopy

NMR experiments were performed on Bruker Avance III spectrometers (Billerica, MA, USA) operating at 14.1 T, equipped with a cryo-enhanced QCI-P probe. All NMR data were collected at a temperature of 288 K where we could observe the maximum number of signals.

### 2.3. Preparation of Samples Used for NMR Assignment Experiments

The native form of ^13^C,^15^N labelled vHVD was prepared in the NMR buffer at concentration of 765 μM with 10% of D_2_O and 100 μM DSS. The denatured form of vHVD was prepared by mixing equal volumes of 850 μM of ^13^C,^15^N labelled vHVD protein and 6 M guanidium chloride (GdmCl) dissolved in the same buffer followed by addition of 10% D_2_O and 100 μM DSS. The final concentrations were 385 μM for vHVD and 2.73 M for GdmCl. All spectra were acquired in 3 mm NMR tubes.

### 2.4. NMR Experiments

3D TROSY type [11,19] experiments for vHVD and vHVD + GdmCl samples were performed to assign the backbone resonances. The data were acquired using 25% NUS sampling schedule. For the 3D HNCO, 4 scans were used on a time domain grid of 1 K × 30 × 80 complex points with spectral width/acquisition time of 12 ppm/141 ms for ^1^H, 30 ppm/16 ms for ^15^N, and 40 ppm/13 ms for ^13^C′. For the 3D HNCOCA and 3D HNCA, 8 scans were used on a time domain grid of 1 K × 30 × 80 complex points with spectral width/acquisition time of 12 ppm/141 ms for ^1^H, 30 ppm/16 ms for ^15^N, and 40 ppm/13 ms for ^13^C^α^. For 3D HNCOCACB and 3D HNCACB, 16 scans were used on a time domain grid of 1 K × 25 × 160 complex points with spectral width/acquisition time of 12 ppm/141 ms for ^1^H, 30 ppm/14 ms for ^15^N, and 80 ppm/12 ms for ^13^C^β^. Overall acquisition time was about 62 h.

H^α^ protons were assigned using a 3D HCACO with sampling schedule comprising 25% NUS and using the parameters: 16 scans on a time domain grid of 1 K × 40 × 64 complex points with spectral width/acquisition time of 12 ppm/141 ms for ^1^H, 40 ppm/6.6 ms for ^13^C^α^, and 20 ppm/21 ms for ^13^C′.

Side chain protons were assigned using 3D HCCCONH and HBHACONH with a sampling schedule using 25% NUS and the parameters: 16 scans on a time domain grid of 1 K × 32 × 128 complex points with spectral width/acquisition time of 12 ppm/141 ms for ^1^H, 30 ppm/17.5 ms for ^15^N, and 12 ppm/17.7 ms for ^1^H. All above mentioned pulse programs were from the Bruker program library (TopSpin3.5).

### 2.5. MUSIC Experiments

The specific parameters for each amino acid selective experiment are reported in the original publication [9]. The modifications performed on MUSIC experiments in this study are described schematically in Figure 1. The additional parameters related to the modification and adjustment are presented in Appendix A. All experiments were performed using nonuniform sampling of 25%, 32 or 64 scans on a time domain grid of 1 K × 200 complex points with spectral width/acquisition time of 12 ppm/141 ms for ^1^H and 30 ppm/109.6 ms for ^15^N for all experiments, except for proline-selected experiments, which used: 1 K × 400 complex points and 60 ppm/109.6 ms for ^15^N.

NMR data were processed by Topspin 4.0.6 (Bruker, Billerica, MA, USA).

The manual assignment was performed using CcpNmr Analysis 2.4.2 (Leicester, UK) [20]. The ^1^H chemical shifts were referred to DSS-d6 in direct dimension, while ^13^C and ^15^N chemical shifts were referenced indirectly.

Random coil chemical shifts of vHVD and vHVD + GdmCl were calculated using POTENCI [21] with neighbor correction and subtracted from the experimental ^1^H^N^, ^15^N, ^13^C′, ^13^C^α^, ^13^C^β^, and H^α^ chemical shifts [21,22]. The chemical shift index (CSI) was calculated according to the original method [23]. Residues with consecutive Δδ^13^C′ or Δδ^13^C^α^ values above 0.7 ppm and below −0.7 ppm indicate alpha helix and beta strands, respectively. The opposite is valid for Δδ^13^C^β^. The CSI for the three nuclei was averaged and reported as “consensus” CSI.

In the figures and in the text, the standard nomenclature for amino acids of the carbon atoms was used, where ^13^C^α^ is the carbon next to the carbonyl group, ^13^C′ is the backbone carbonyl group, and ^13^C^β^ is the sidechain carbon next to ^13^C^α^ [24].

Data availability. The final chemical shift assignments list of the backbone ^1^H^N^, ^15^N, ^13^C^´^, ^13^C^α^, ^13^C^β^, and H^α^ and H^β^ atoms of vHVD and vHVD + GdmCl at 15 °C have been deposited in the Biological Magnetic Resonance Data Bank (http://www.bmrb.wisc.edu/) under BMRB accession code 50593. Other data and pulse sequences are available from the corresponding authors upon reasonable request.

## 3. Results

### 3.1. Selective Detection of Different Types of Amino Acids in vHVD + GdmCl Spectra

As a denaturing agent, we used guanidium chloride, see introduction and materials and methods. In the TROSY ^1^H-^15^N HSQC spectrum of vHVD with 2.73 M GdmCl (vHVD + GdmCl) (Figure 2A) all expected backbone amide signals of vHVD according to the sequence (Figure 2 top) are observable and importantly have narrow linewidths (<14 Hz for ^1^H) compared to native vHVD where some cross peaks were weak or even below detection limit. In addition to the favorable relaxation properties of the fully unfolded vHVD, it was also assumed to have similar coupling constants in the different amino acids and thus expected to have optimal conditions for the MUSIC experiments [9] adopted and modified in this study (Methods and Figure 1) providing increasing sensitivity and resolution.

Indeed, in Figure 2A, as an example, the ^1^H-^15^N HSQC spectrum of vHVD + GdmCl is superimposed with spectra of MUSIC experiments showing sequential, NH(S + 1), and own differently labelled NH(S) selected cross peaks. Importantly, without MUSIC spectra with increased resolution, comparable to the ^1^H-^15^N HSQC spectrum, it would be impossible to directly discriminate between cross-peaks in crowded areas. This is illustrated in expanded part of spectrum presented in boxes below Figure 2A.

Figure 2B shows the changes of intensity of the cross peak of 438G presented as projections through this peak depending from the applied Gly-MUSIC type experiments: (a) with C^α^ chemical shift evolution in t_1_ [9] (Figure 2Ba); (b) with sensitivity enhancement increasing intensities √2 times as expected [12] (Figure 2Bb); (c) with CO chemical shift evolution in t_1_ [9] (Figure 2Bc); (d) with sensitivity enhancement increasing intensities √2 times as expected [12] (Figure 2Bd). These data show that for IDPs with favorable relaxation properties, the S/N ratio could be increased by up to 2.2 times using option (d).

In Figure 2C, as an example, the ^1^H-^15^N HSQC spectrum of vHVD + GdmCl is superimposed with spectra of MUSIC experiments showing the next neighbor, i + 1, to the selected, i, residue type of amino acid cross peaks. The peaks are differently labelled for different types of amino acids: NH(S + 1), NH(A + 1), NH(D + 1), NH(E + 1) and N(R + 1) amino acids. Finally, a comprehensive sets of data for NH(T + 1), NH(I + 1), NH(G + 1), NH(N + 1), NH(Q + 1), and NH(L + 1), including NH(T), NH(I), NH(G), NH(S), NH(L), NH(A), NH(D), NH(E), and N(R), were collected, and corresponding cross peaks in the ^1^H-^15^N HSQC spectrum of vHVD + GdmCl were summarized to the different classes of amino acids for discrimination purposes.

### 3.2. Proline Assignment

A specific feature of IDPs is that they often have stretches of repetitive sequences and they are rich in proline residues. The vHVD belong to a family of proline-rich IDP proteins containing 25 prolines (Figure 2 top marked by red), corresponding to about 13% of all aa, which are interspersed along the amino acid sequence and are likely to have an impact on its structure and function. They are combined in four different motifs: (1) single, X^1^pX^2^, or (2) double, X^1^ppX^2^, prolines surrounded by more than one aa; (3) two prolines separated by one aa, X^1^pX^2^pX^3^, flanked by more than one other aa; (4) three prolines separated by one aa, X^1^pX^2^pX^3^pX^4^, flanked by more than one other aa; and (5) ppX^1^pX^2^pX^3^ where X^1^, X^2^, and X^3^ are different nonproline aa in the vHVD sequence. Generally, this imposes an additional complication for the sequence-specific resonance assignment of IDPs because proline residues are break points in the sequential assignment walk. By increasing resolution in MUSIC experiments in the vHVD + GdmCl sample, we identified almost all correlations through one proline in motifs (1), (3), (4), and partly (5) with the addition of the aa being terminal to i-1 or i + 1 proline in motif (2) and (5) (Figure 3).

Additionally, aa(P-1) and aa(P + 1) were identified through the superposition of selective MUSIC proline spectra vs. different types of aa MUSIC spectra (Figure 2) and used as starting or reference points for assignment. Based on these data as well as data related to the type of aa for cross peaks in the ^1^H-^15^N HSQC spectrum of vHVD + GdmCl, the manual sequential assignment of the backbone resonances of vHVD + GdmCl was easily performed in a reasonably short time, achieving up to 95–99% by following the connectivity (^1^H^N^_i_/^15^N_i_, ^1^H^N^_i_/^13^C^O^_i−1_, ^1^H^N^_i_/^13^C^α^_i−1_, and ^1^H^N^_i_/^13^C^β^_i−1_) based on a combination of a set of high-resolution 25%NUS type 3D TROSY type correlation experiments.

### 3.3. Assignment of the ^1^H and ^15^N in vHVD by Back-Titration

After assigning the back-bone and side chain resonances in unfolded vHVD + GdmCl, the next step in our approach was to extrapolate the assignments of ^1^H and ^15^N resonances to native vHVD by back titration reducing the amount of GdmCl through dilution of the sample by buffer and keeping the concentration of protein the same. It was enough to obtain 5 titration points to be able to unambiguously follow the trace of movement of the ^1^H-^15^N cross peaks of individual aa in the ^1^H-^15^N HSQC spectrum (Figure 4A). Four regions were chosen and expanded as examples showing the clear traces of chemical shift perturbation of the cross peaks (Figure 4Aa–d). In Figure 4A, the trace showing the cross peaks of 484S became nonobservable in native vHVD.

Extrapolated assignment of ^1^H, ^15^N data of vHVD, and of ^13^C resonances vHVD + GdmCl were combined and copied to the vHVD experimental data set in ccpn assignment module for final verification of the assignment of vHVD. Discussion about differences in carbon resonances is done below.

The overall time spent for backbone assignment of vHVD was estimated to be about 2 weeks, which includes the time spent for NMR data collection for the two data sets of vHVD and vHVD + GdmCl and time for assignment and follow up verification. Using this approach, we were able to assign 91.5% ^1^H(N), 90.5% ^15^N_i_, 92.1% ^13^C^α^, 91.2% of ^13^C^β^, and 79.4% of ^13^C′ signals for vHDV + GdmCl and 90.9% ^1^H(N), 89.9% ^15^N_i_, 94.2% ^13^C^α^, 89.0% of ^13^C^β^, and 93.4% of ^13^C′ signals for native vHDV.

## 4. Discussion

### 4.1. Chemical Shift Assignment Strategy

Backbone sequential resonance assignment of large IDPs is so far still a challenge. The development of TROSY type [11,19], BEST-TROSY [25,26], and the nonuniform sampling (NUS)-based NMR techniques [27,28,29] has made it possible to enhance the resolution of multidimensional NMR spectra and to study larger IDPs by NMR [30]. Recently, we performed an evaluation of the manual traditional approaches vs. automated, i.e., NUS-based targeted acquisition procedure (TA) [28,31,32,33] as a mean to decrease the acquisition and analysis time for larger IDPs [30]. It has been demonstrated that in application for IDPs, TA procedure using BEST-TROSY [25] type experiments: 3D HNCO, 3D HNCOCA, 3D HNCA, 3D HNCACO, 3D HNCOCACB, and 3D HNCACB can be preferable from an NMR time-consuming point of view to traditional TROSY type experiments, leading to significantly decreased acquisition time. Indeed, for the IDP CHIKV HVD (about 230 aa) with a sample concentration of about 0.3 mM in a 3 mm NMR tube, all sets of TA BEST-TROSY experiments performed on a 700 MHz spectrometer equipped with a 5 mm CP-TCI probe were executed in only 24 h, which traditionally takes a few days [30].

Nevertheless, the nonconventional processing NUS-collected TA data and automatic assignment procedure of large IDPs is still a challenge and is not readily accessible to the NMR community. Moreover, in the new algorithm proposed by us of IDP backbone assignment [30], clean peak lists provided by the TA procedure were mostly automatically assigned by FLYA software [34] but needed manual follow-up verification with CARA software [35] using high-resolution 3D spectra that were also provided by the TA. Additionally, the TA approach was also tested by us through combining automated 5D HACACONH with XLSY approach [36]. The peak list from the 5D HACACONH spectrum was used as an independent reference and control for the curated TA assignment as well as a source of ^1^H^α^ protons. Recently, a new set of H^α^-detected and H^α^-start, i.e., H^N^-detected 4D experiments [37,38] were proposed to improve the efficiency in assignment.

Despite that the proposed data analysis resulted in high confidentiality of the IDP assignment, it is urgent to find more simple protocols easily usable for NMR users without applying additional time-consuming multidimensional NMR experiments and using conventional readily available processing software.

The application of MUSIC type experiments is mainly limited to small-sized FPs due to the unfavorable relaxation properties of large FPs. This is not the case for IDPs.

One of the ways to detect hidden residual structure was recently demonstrated by applying chemical denaturation titration (CDT)-NMR [13], in which chemical denaturants disrupt residual structure and the rapid exchange between conformers in IDP ensembles providing good-quality relaxation properties and spectra.

The vHDV IDP studied in this work is a 189-aa-long fragment (Figure 2) with repetitive short and long aa sequence parts, rich in proline residues and one of the largest IDPs studied so far. Broadening of many amide H and N(H) resonances in vHDV below the detection limit due to different types of interactions, molten globule-like behavior, and slow exchange process make application of the well-established NMR techniques, based on the ^1^H-^15^N amide HSQC experiment, less effective and limit the completeness of assigned backbone sequential resonances.

Keeping this in mind, we propose the inverse-step assignment protocol. First, we have performed assignment of the backbone and side chain resonances of the vHDV in manual manner. In order to achieve a maximum possible unfolding of the protein, GdmCl was added. Due to favorable relaxation properties leading to high quality of the spectra of the unfolded vHDV + GdmCl, traditional 3D amide proton-observed TROSY type experiments with NUS data collection were used for the assignment. The obtained data were complimented with a set of short 2D selective sensitivity that improved HSQC/TROSY modified, semiconstant type MUSIC experiments. Those experiments provided a higher resolution in the ^15^N indirect detection, and we could distinguish sets of cross peaks belonging to different types of aa. Noteworthy, that as an alternative to manual assignment, the automated assignment protocol TSAR [38] (based on 3D experimental data collected in this study with assignment of vHDV + GdmCl about 60% aa) or the FLYA assignment [34] also provides good results but needs more extensive verification and follow-up manual assignment [30]. Additionally, for our sample, inclusion of the experimentally obtained type of aa to these automatic assignment protocols only slightly improved the prediction (data not shown and will be presented elsewhere).

Next, the assigned cross peaks in ^1^H-^15^N HSQC type experiments of unfolded vHDV + GdmCl were back-extrapolated through dilution of vHVD to zero GdmCl concentration as presented in Figure 4. Noteworthy, the cross peaks missing in the native vHDV spectrum were already observed in the presence of 0.2 M of GdmCl. The list of amides ^1^H and ^15^N for vHDV provided by this procedure was combined with the assigned ^13^C^α^, ^13^C^β^, and ^13^C′ of vHDV + GdmCl and refined to a final assignment using ccpn software.

### 4.2. Secondary Structure of Native vHDV

One of the main approaches to extract information related to the presence of any structural elements in IDP is to estimate the level of disorder based on the chemical shift index (CSI) for resonances ^1^H(N), ^15^N, ^13^C^α^, ^13^C^β^, ^13^C′ and ^1^H^α^ for every aa and comparing it with a reference table of the corresponding resonances for unfolded proteins [22]. Despite that this approach is commonly accepted as a powerful and valid approximation of secondary structure of IDP, there are still only a few reference data sets available, which could show some variance between each other and lead to possible ambiguities. It is known that there are deviations and exceptions in data sets depending on the aa sequences and pH [22,39,40,41].

In this study, we obtained data sets of assigned resonances of the sidechain and back-bone resonances of the vHDV protein, which possibly is heterogeneous under experimental conditions, as well as its counterpart obtained by denaturation of vHDV by adding GdmCl. The later, can play a very important role as a convenient reference due to the expected random coil values. By this way of comparison, the problem of sequence dependence ambiguities could be cancelled. In Figure 5, super positions of the chemical shift deviations (CSD) between ^1^H(N), ^15^N_i_, ^13^C^α^, ^13^C^β^, ^13^C′, and ^1^H^α^ resonances of vHDV (Figure 5 red bars) and vHDV + GdmCl (Figure 5 black bars) vs. POTENCI data base library are presented. Preliminary reviewing of the CSD values for both proteins shows that they are generally lower or similar to the threshold used as secondary structure criteria [42]. Nevertheless, it is evident that CSD values in regions 440–455 and 504–510 aa are reduced in the vHDV + GdmCl sample vs. native vHDV. To amplify this effect in Figure 6, we presented Δδ^13^C^α^–Δδ^13^C^β^ values that are commonly used for reporting secondary chemical shift with the advantage of cancelling out chemical shift referencing errors [41]. For native vHDV, there are four regions with positive values indicating an α-structure tendency (Figure 6 top). One region between aa 504–510 has consecutive values of about +/−2 ppm, which can be interpreted as having a fully formed secondary structure. Indeed, even in the presence of denaturing reagent, vHDV + GdmCl still displays a tendency towards α-helix propensity in C-end of protein between aa 505 and 510 (Figure 6 bottom). Three other regions located between aa 362–366, 421–432, and 442–454 (Figure 6 top) have lower positive values indicating partially formed structures and are not observed for vHDV + GdmCl (Figure 6, bottom). Importantly, there are two regions, i.e., aa 372–378 and 411–415, with negative lower values (Figure 6 top) in vHDV, which are not observed in vHDV + GdmCl (Figure 6 bottom), indicating a tendency of forming a β-strand with low population. The conclusion of an α-helix formation between aa 505–510 based on the Δδ^13^C^α^–Δδ^13^C^β^ values is in line with data for differences of Δδ^13^C′ and Δδ^1^H^α^ chemical shifts from those found in a random coil conformation (Figure 5) where Δδ^1^H^α^> −0.1 and Δδ^13^ C′ > 1.0 for both vHDV and vHDV + GdmCl. Noteworthy, Δδ^1^H(Ν) consecutive values for vHDV + GdmCl (Figure 6) show a large positive tendency compared to vHDV, which can be due to the additional contribution of interaction between amide protons and GdmCl.

The nsP3 HVDs of chikungunya virus (CHIKV) and Venezuelan equine encephalitis virus (VEEV) have no homology but are both involved in replication complex assembly and function. Previously, we have shown that nsP3 HVD CHIKV is completely disordered with low tendency to form secondary structures in free form [30,43]. Despite this, we have reported the possibility of specific interactions of nsP3 HVD CHIKV with the N-terminal part of CD2AP, containing three different SH3 domains and its individual SH3 domains [30]. In this new study, based on NMR data, we showed that nsP3 HVD protein of VEEV is also mostly disordered, but has several fragments with tendency to form secondary structures. The most profound secondary structure is a short stable α-helix in its C-terminal fragment, which is predicted to mediate interaction with the members of the Fragile X syndrome protein complex [16,18]. It would be interesting to define the structure of the full C-terminus of VEEV HVD, which contains two almost identical repeats. The availability of fully assigned spectra of VEEV HVD will allow us to identify the biding sites for all interacting host factors and study their roles in virus replication.

## 5. Conclusions

To conclude, we propose a robust and reasonably quick protocol to assign the backbone and sidechain resonances in NMR spectra of IDPs. It assumes a twostep process: first, assignment of the unfolded protein in the presence of denaturant, GdmCl. The significantly improved quality of the NMR spectra allows the use of a combination of experiments defining the type of aa and accelerated data acquisition by NUS of H(N)-type 3D conventional experiments. To make MUSIC type experiment more efficient, we made modifications to the available pulse sequences with (a) semiconstant time functions to boost resolution in the ^15^N dimension (b) with TROSY and (c) sensitivity improvement options. Second, back titration from the unfolded protein to its native form allows the transfers of amide resonances assignments. This protocol was applied to the structural study of nsP3 HVDs of Venezuelan equine encephalitis virus (VEEV). Using this protocol two data sets of chemical shifts belonging to the native protein, vHVD, and its unfolded form, vHVD + GdmCl, were obtained allowing detailed analysis of its secondary structure and revealing its structural heterogeneity in solution.

## Figures and Tables

**Figure 1 molecules-25-05824-f001:**
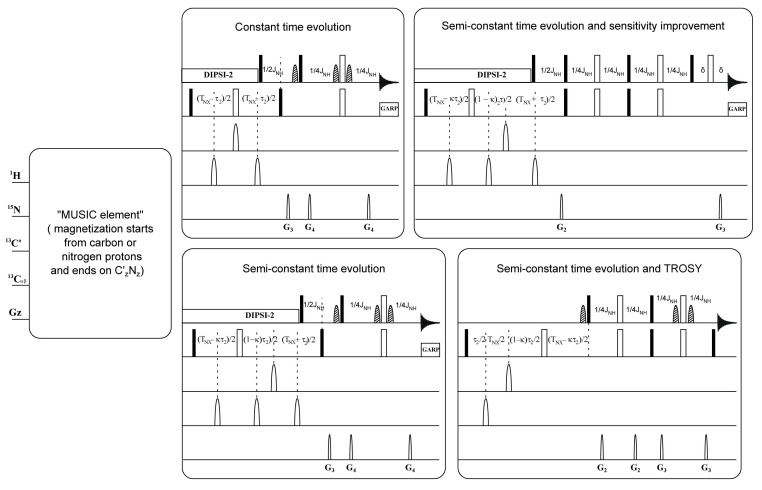
Schematic representation of MUltiplicity Selective In-phase Coherence (MUSIC) pulse sequences and the applied modifications. The final block of standard Bruker library programs is adopted as semiconstant time evolution. Here, 90° and 180° rectangular pulses are represented by filled and unfilled bars, respectively. Further, 90° water selective pulses are represented by dashed shapes. Moreover, 180° ^13^C pulses are represented by unfilled shapes. Other details are given in the original publication [9] and those relating to the modification and adjustment performed in our experiments are given in Appendix A.

**Figure 2 molecules-25-05824-f002:**
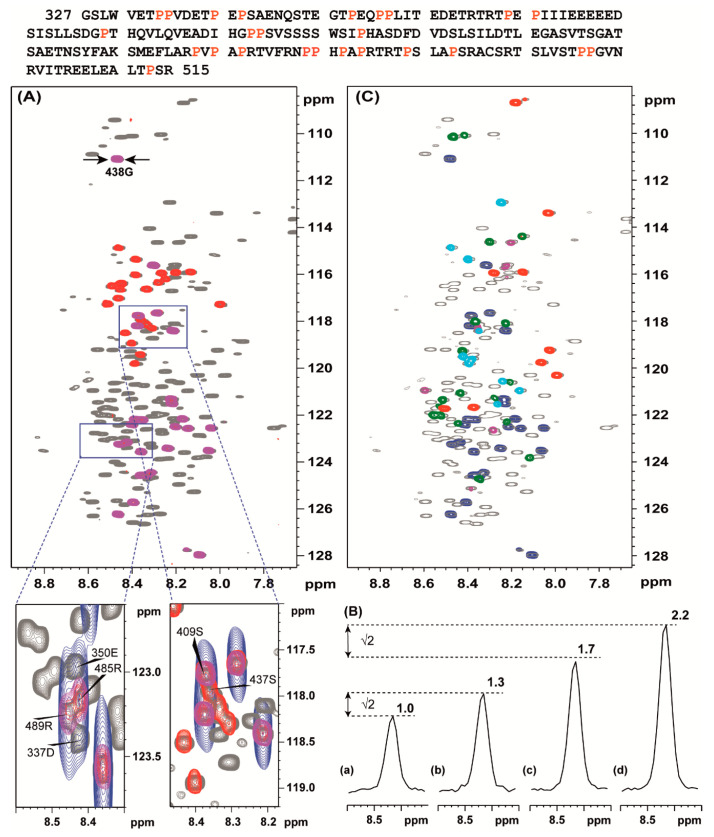
Top: the aa sequence of Venezuelan equine encephalitis virus (VEEV) nonstructural protein 3 (nsP3) hypervariable domains (HVD), VEEV HVD (vHVD). Prolines are labelled in red. Superposition of the ^1^H-^15^N HSQC of vHVD + GdmCl (grey) with MUSIC type of experiments performed with acquisition time 0.109 s in ^15^N direction giving a resolution of 9.1 Hz showing: (**A**) (magenta) S + 1 and (red) S aa, (**C**) (red) D + 1, (blue) S + 1, (green) E + 1, (cyan) A + 1, and (magenta) R + 1 amino acids. The part of the spectra in (A) surrounded by boxes are presented below showing (blue) corresponding MUSIC spectrum acquired traditionally as constant time experiment in ^15^N direction with a resolution of 45.6 Hz, acquisition time 0.022 s. (**B**) The cross-sections through cross peak 438 G (shown in panel (**A**)) for 4 types of MUSIC pulse sequences: (**a**) intensity in the traditional MUSIC experiment is set to 1; (**b**) in the sensitivity enhance MUSIC experiment (Figure 1) corresponding intensity is increased by √2 times as expected [12]; (**c**) in the traditional MUSIC experiment but with C=O chemical shift evolution, the intensity increased 1.7 times, and (**d**) in the traditional MUSIC experiment but with C=O chemical shift evolution, sensitivity enhanced by √2 times.

**Figure 3 molecules-25-05824-f003:**
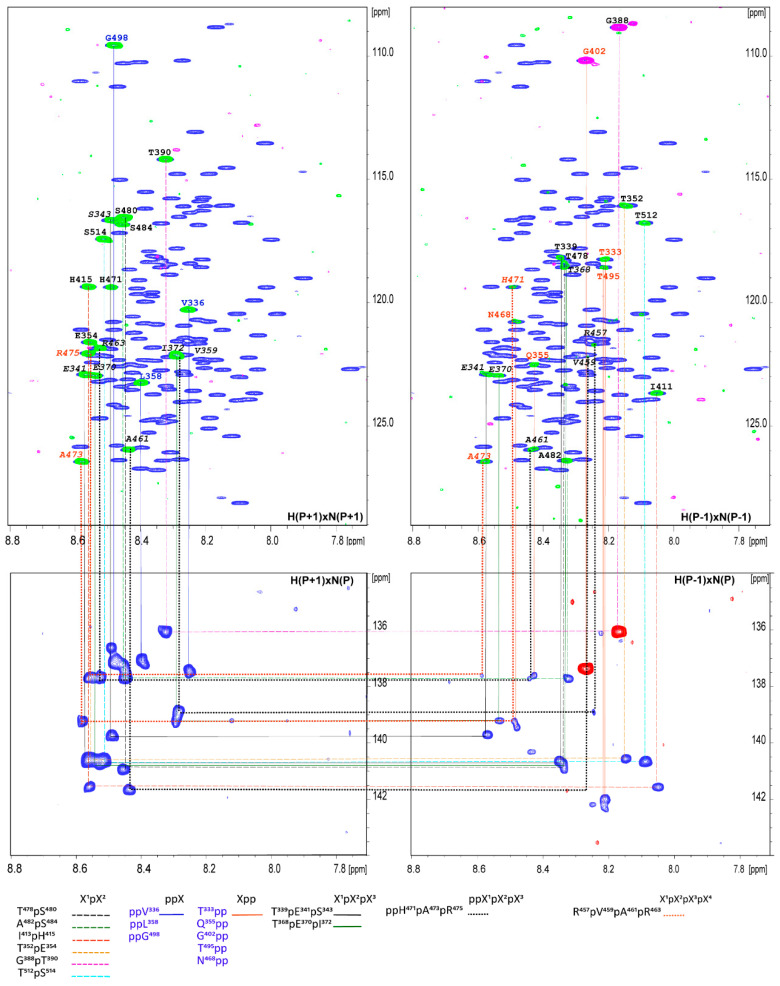
Proline connectivity in vHVD + GdmCl in the four types of proline aa selected in MUSIC type of experiments have been presented and indicated by color and type of line according to different types of connectivity defined in below panels. The type of experiment is marked in the corners of the panels. The assignment of the aa of connected cross peaks is shown near cross peaks according to the sequence number.

**Figure 4 molecules-25-05824-f004:**
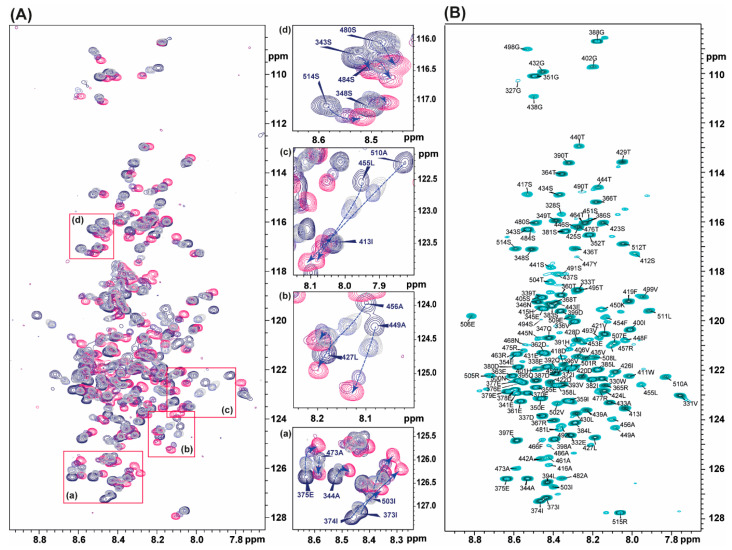
(**A**) Titration of vHVD by GdmCl. Left: superposition of the ^1^H-^15^N TROSY spectra of native vHVD (navy), vHVD + 0.2 M GdmCl (mid-grey), vHVD + 0.4 M GdmCl (mid-blue), vHVD + 1.3 M GdmCl (mauve), vHVD + 1.5 M GdmCl (purple), and vHVD +2.7 M GdmCl (pink). The expanded part of spectrum in boxes (**a**–**d**) are presented on right site of Panel where by arrows (blue dash lines), the trajectories of the chemical shift perturbation (CSP), are showed for the selected cross peaks labelled by bold font. (**B**) ^1^H-^15^N HSQC of native vHVD with assignment.

**Figure 5 molecules-25-05824-f005:**
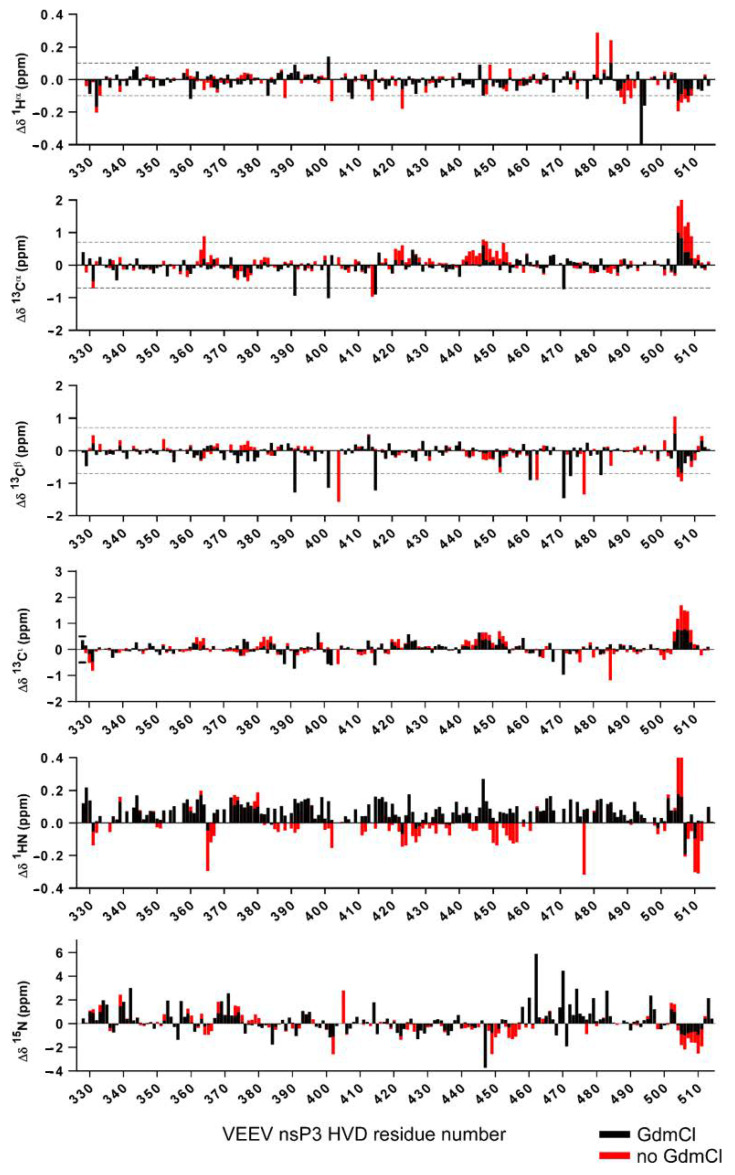
Superposition of the chemical shifts deviations from the random coil chemical shift calculated by POTENCI [21] for ^1^H(N), ^15^N_i_, ^13^C^α^, ^13^C, and ^13^C′ nuclei of native vHDV (red bars) vs. vHDV + GdmCl (black bars) are presented.

**Figure 6 molecules-25-05824-f006:**
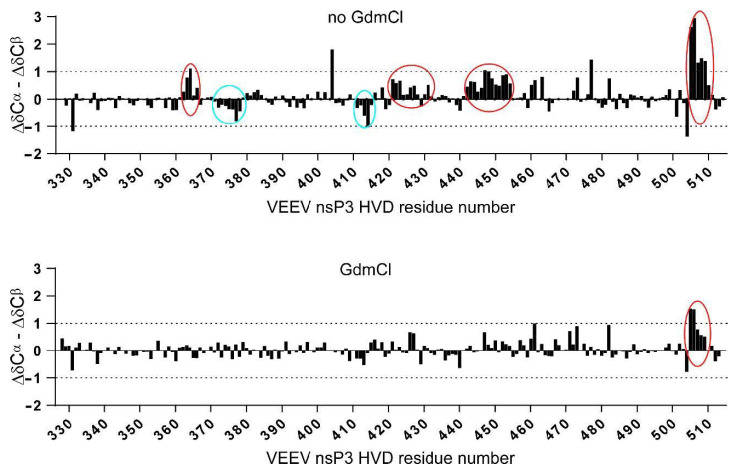
ΔδC^α^–ΔδC^β^ secondary chemical shifts for native vHVD (top) and vHVD + GdmCl (bottom). Positive values showing inside in 4 red circles indicate α-structure tendency, and negative values showing inside 2 blue circles indicate β-structure tendency. Addition of denaturing reagent (vHVD + GdmCl) indicates that this protein adopts mostly an extended coil-like structure but still displays a tendency toward α-helix propensity in C-terminal of protein, aa 505–510.

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
