# Peer review of "Novel NMR Assignment Strategy Reveals Structural Heterogeneity in Solution of the nsP3 HVD Domain of Venezuelan Equine Encephalitis Virus"

_molecules, 2020, doi:10.3390/molecules25245824_

Round 1
Reviewer 1 Report
In this manuscript Agback et al. provided a new strategy of sequential assignments for intrinsically disordered proteins (IDPs). This strategy combines traditional HN-detection 3D NMR spectra and improved MUSIC pulse sequences. Authors also presented usefulness of chemically denatured IDP for initial stage of the assignments which could be traced to NMR signals of undenatured IDP using titration methods. This new method was successfully applied to assign NMR signals of the nsP3 hypervariable domain of Venezuelan equine encephalitis virus.
Authors modified MUSIC pulse sequence by using TROSY sequences and semi-constant time evolution period, which improved the resolution of indirect 15N dimension. This higher 15N resolution was very useful for authors to assign amino acid residue types and residues flanking prolines. Particularly, it helped sequential assignments of prolines, which is a common problem for both IDPs and structured proteins.
Authors clearly demonstrated technical advantages of their new NMR experiments by showing spectra and simulation results. Overall, this is a well written manuscript with technically sound experimental data which will help researchers in the IDP field.
Minor comments:
Why authors deleted the last 35 aa of natural HVD?
Fig. 3 Font size used in the figure is too small to read.
It appeared that all NMR spectra were taken at 288K (15°C). Is there any change of intrinsic secondary structures of nsP3-HVD between 15°C and 37°C?
Lines, 130, 141, 142, 181 … there seem to be errors of Greek fonts.
Reviewer 2 Report
In this manuscript, Agback et al introduced a strategy for specific resonance assignment of large IDPs based on traditional 3D experiments, chemical denaturation and optimization of the MUSIC pulse sequences. They apply this protocol to the study of vHVD from VEEV and found some possible structured regions inside this Intrinsically disordered protein (IDP). Although none of these methods in the protocol can be considered “novel” (separated), the combined strategy is interesting, indeed. The possibility of significant contributions to the “unstructured biology” field might be one of the greatest strengths of modern protein solution-NMR, filling the holes left by the recent developments in Cryo-EM and protein crystallography. NMR needs to fulfil the IDP field with enough structural and dynamic data, especially because it is the only technique capable of doing that with high-resolution. Therefore, new protocols to deal with the still-challenging assignment of large IDPs are much welcomed, especially for the proline-rich proteins and regions. However, I have some issues especially to the structural heterogeneity of vHVD in solution and the level of discussion. I believe this is important to be addressed since it is one of the main findings described in the paper. I recommend the paper for publication after revision.
Majors:
- Is there any reason for the absence of the last 35 aa of natural vHVD? Since the C-terminal region of the vHVD mediates interaction with the members of cellular Fragile X syndrome 26 protein family, and the finding of this short, and stable, α-helix in its 25 C-terminal fragment is one of the main novelties of the paper, it should be more well-discussed.
- Related to the possible structured regions, I believe that their presence and how they are disturbed should be analysed by a technique like CD, exploring not only the hypothesis for the presence of those partially formed “structured-regions”, but also, the effect of GdmCl using a traditional titration experiment (doi:10.1038/nprot.2006.229) and, even more important, of the protein reversibility in this refolding strategy. IDPs are prone for disorder-to-order transitions, especially when minor changes in the physicochemical properties of the medium are introduced. Since the concentrations naturally used in NMR experiments are high, I still do not discard the possibility of an induced folding based on a weak self-interaction. If the authors disagree, the reasons should be discussed.
- There are some disorder predictors, in particular, the PONDR® VL-XT (http://www.pondr.com/) well designed for the prediction of disordered regions longer than 8 aa. I tested the sequence given in figure 2 and it returns some regions predicted to be “ordered” that correlates with the CSD data. I think the authors could include this prediction to enhance the level of discussion. Just remember that the output for the XT predictors starts at the first or last sequence position and continues for 14 residues inward from the termini, so a longer protein sequence than the one presented in figure 2 should be used.
- This is more curiosity and, maybe, a suggestion for the future than an issue itself. GdmCl is a strong chaotropic agent but it is charged and can interfere with the denaturation process by unspecific binding due to electrostatic interactions. This can also affect the refolding strategy if the interaction is strong. Besides, it decreases the cryoprobe efficiency due to increased sample conductivity. Why not urea, instead of GdmCl, for the back-titration strategy?
Minors:
- line 33, it should be “structured proteins” or, even better, “well-structured proteins”
- The colours used in figure 4 made it difficult for me to do the tracking of the resonance lines along with the titration. I suggest a change in colours that are less alike by eyes.
- In Figure 5, it should be GdmCl instead of GuCl
- In line 361, GdmCl is not considered a detergent
- In line 406, “assignment of the fully disordered protein” should be “assignment of the fully unfolded protein”, or something similar. Disordered structures and unfolded structures are not the same. While the disorder is an intrinsic property, unfolded structures are induced by an external agent, and they differ in some physical properties like Rg, the end-to-end distance of the molecular chain, CD spectrum, besides others.
- There are some minor grammatical errors along with the text, especially in the discussion section.
Reviewer 3 Report
This manuscript used novel NMR approaches to assign the spectra for the nsP3 HVD of Venezuelan equine encephalitis virus (VEEV). The HVDs of CHIKV and VEEV have no homology but are both involved in replication complex assembly and function. The author has found that VEEV nsP3 HVD is also mostly disordered but contains a short stable -helix in its C-terminal fragment, which mediates interaction with the members of cellular Fragile X syndrome protein family. This manuscript obtained allowing detailed analysis of its secondary structure and revealing its structural heterogeneity in solution.
The manuscript has a certain innovative, but there are still some problems inadequacies should be improved, such as:
- Materials and Methods: I suggest to change Line 126 “Samples used for NMR assignment experiments” to “Preparation of samples for NMR distribution experiments”.
- Fig.5 and 6: I suggest that the author change the font "VEEN NSP3 HVD Residue Number" in the picture.
- Line 98: The caption font is inconsistent with the others.
- There are many problems with the typesetting of the manuscript. For example, there are a lot of blanks in the sixth and seventh pages. There should be no spaces in many parts of the manuscript. The Line spacing of Line 123-125 is too large.
- The pictures in the manuscript are not clear. It is suggested that the author improve the sharpness of the pictures
- There are too many grammatical errors.
- There are too few references in the manuscript , so it is suggested that the author add more references
